# Physiological and Comparative Transcriptome Analyses of the High-Tillering Mutant mtn1 Reveal Regulatory Mechanisms in the Tillering of Centipedegrass (*Eremochloa ophiuroides* (Munro) Hack.)

**DOI:** 10.3390/ijms231911580

**Published:** 2022-09-30

**Authors:** Ling Li, Chenming Xie, Junqin Zong, Hailin Guo, Dandan Li, Jianxiu Liu

**Affiliations:** The National Forestry and Grassland Administration Engineering Research Center for Germplasm Innovation and Utilization of Warm-Season Turfgrasses, Institute of Botany, Jiangsu Province and Chinese Academy of Sciences, Nanjing 210014, China

**Keywords:** centipedegrass, tillering, *mtn1* mutant, RNA-Seq, hormone, starch and sucrose

## Abstract

Tillering is a key factor that determines the reproductive yields of centipedegrass, which is an important perennial warm-season turfgrass. However, the regulatory mechanism of tillering in perennial plants is poorly understood, especially in perennial turfgrasses. In this study, we created and characterised a cold plasma-mutagenised centipedegrass mutant, *mtn1* (*more tillering number 1*). Phenotypic analysis showed that the *mtn1* mutant exhibited high tillering, short internodes, long seeds and a heavy 1000-seed weight. Then, a comparative transcriptomic analysis of the *mtn1* mutant and wild-type was performed to explore the molecular mechanisms of centipedegrass tillering. The results revealed that plant hormone signalling pathways, as well as starch and sucrose metabolism, might play important roles in centipedegrass tillering. Hormone and soluble sugar content measurements and exogenous treatment results validated that plant hormones and sugars play important roles in centipedegrass tiller development. In particular, the overexpression of the auxin transporter *ATP-binding cassette B 11* (*EoABCB11*) in Arabidopsis resulted in more branches. Single nucleotide polymorphisms (SNPs) were also identified, which will provide a useful resource for molecular marker-assisted breeding in centipedegrass. According to the physiological characteristics and transcriptional expression levels of the related genes, the regulatory mechanism of centipedegrass tillering was systematically revealed. This research provides a new breeding resource for further studies into the molecular mechanism that regulates tillering in perennial plants and for breeding high-tillering centipedegrass varieties.

## 1. Introduction

Centipedegrass (*Eremochloa ophiuroides* (Munro) Hack.) is a perennial warm-season C_4_ grass originating from China with wide adaptability, high barrenness and acid tolerance [1,2]. It has great potential use for soil conservation and environmental remediation and as a forage grass [3]. As a perennial turfgrass, centipedegrass is propagated by stolons and seeds, which directly depend on the continuous tillering of stolon nodes [4,5]. However, the regulatory mechanism in the development of tillering in centipedegrass has not been reported.

Tillering is an important breeding trait that determines reproduction efficiency and yields in perennial plants. It appears from the node and grows independently of the mother stem from its own roots [6,7]. Previous studies have indicated that auxin (IAA), gibberellic acid (GA) and cytokinin (CTK), are the most important factors affecting tillering [7,8,9,10]. Auxin has been reported to inhibit tillering. In rice, the overexpression of *PIN-FORMED1* (*OsPIN1), OsPIN2* and *OsPIN9* increases the amount of tillering [8,11]. In Arabidopsis, the *atiaa17* mutant shows a high branching phenotype [12]. GA is involved in the regulation of tillering, but its effect on tillering (branch) is different in different species. Previous studies found that GA might promote tillering/branch in perennial woody plants, such as sweet cherry [13] and *Jatropha curcas* [14], but inhibit tillering/branch in annual herbaceous plants, such as tall fescue [15], orchardgrass [16] and tomato [17]. Previous studies confirmed that CTK improves tillering development [15,18]. In rice, the overexpression of the CTK degradation-rated genes *Cytokinin oxidases/dehydrogenases 4* (*OsCKX4*) and *Zeatin O-glucosyltansferase 1* (*OsZOG1*) decreases tillering numbers [18,19]. In the oilseed rape shoot apical meristem mutant *dt*, the *Lonely guy 6* (*LOG6*) expression level is upregulated [20].

Sugar also plays an essential role in tillering. Sugar is a signalling regulator that interacts with other signal transduction factors to regulate tillering [21,22,23]. Tillering is closely correlated with sugar transport, metabolism and signalling [24,25]. In Arabidopsis, the overexpression of the *Trehelaose 6-phosphatase* (*TPP*) gene obviously decreases the branch number [24]. In maize, inflorescence branching is increased by mutations in the *ZmTPP4* gene [25]. Otherwise, it has been found that sugar availability antagonises the auxin-induced repression of bud outgrowth in peas and roses [21,26].

The ATP-binding cassette B (ABCB) subfamily is the second largest ATP-binding cassette (ABC) protein subfamily and is involved in the transmembrane transport of hormones, metals, iron, etc. Some ABCB transporters are involved in tillering (branching) by regulating the transport of auxin [27,28]. Arabidopsis *atabcb19* and *atabcb1abcb19* mutants show lower IAA content and high branching [28].

Reproduction and yields are directly affected by tillering in perennials [1,6]. However, the molecular mechanisms of tillering in perennial plants are poorly understood. Mutants with high or low tillering abilities are ideal materials for studies of the molecular mechanisms of tillering. In this study, phenotypic character and transcriptomic analyses were first performed on a high-tillering mutant, *mtn1* (*more tillering number 1*). Then, candidate gene expression, endogenous plant hormone content measurement and exogenous hormone treatment were conducted to verify the transcriptomic results. Otherwise, the ABCB transporter *EoABCB11* was selected from the differentially expressed genes (DEGs), and its effects on tillering were verified in transgenic Arabidopsis. Finally, SNPs were identified. Our study will provide support for further exploration of the mechanisms underlying tillering and provide potential material for the breeding of high-tillering turfgrass cultivars.

## 2. Results

### 2.1. Morphological Analysis of the mtn1 Mutant

Compared with those of the WT, the primary and secondary tillering numbers of mtn1 increased by 88.0% and 125.9%, respectively (Figure 1A,B and Table 1). Tiller bud number showed no significant difference between *mtn1* mutant and WT (Table 1). The first, second and third tiller bud lengths of mtn1 were higher than those of WT, which increased by 38.24%, 76.51% and 48.14%, respectively (Figure 1C and Table 1). The internode length of *mtn1* was shorter than that of WT, while the internode diameter showed no significant difference between *mtn1* and WT (Figure 1D and Table 1). The shoot and root dry weights of *mtn1* increased by 77.89% and 33.38%, respectively, compared with those of WT (Table 1). Both seed length and 1000-seed weights in *mtn1* were higher than those in WT, which increased by 17.62% and 26.26%, respectively (Figure 1E and Table 1). However, seed width showed no significant difference between *mtn1* and WT (Table 1).

### 2.2. RNA-Seq Data Analysis

A total of 279.00 million raw reads were obtained from the six libraries, and 96.08% of the reads were considered clean reads (Appendix A). The GC contents ranged from 53.93% to 54.95%, Q20 (proportion of nucleotides with a quality value greater than 20 in clean reads) ranged from 98.43% to 98.60% and Q30 ranged from 95.34% to 95.69% (Appendix A). As a result, 93.70% to 94.90% of the total clean reads and 87.42% to 89.30% of the unique reads were mapped to the centipedegrass genome (Appendix A). Among them, 81.52% to 83.46% of clean reads were mapped to exonic regions, 4.34% to 5.04% were mapped to intronic regions and 12.10% to 13.48% were mapped to intergenic regions (Appendix A).

### 2.3. Identification of DEGs between the mtn1 Mutant and WT

To explore DEGs involved in high tillering, a comparison was performed between the *mtn1* mutant and WT. A total of 3025 DEGs were detected, including 1589 upregulated genes and 1436 downregulated genes (Figure 2A,B). The expression patterns of the DEGs were hierarchically clustered (Figure 2C).

### 2.4. Functional Analysis of DEGs

According to gene ontology (GO) subcategories, the DEGs were divided into three main processes: biological processes (BP), cellular components (CC) and molecular functions (MF) (Appendix A). Some DEGs were annotated with more than one GO term. The small molecule metabolic process, ion transport and response to stress were highly enriched among the biological processes. The top three highly enriched processes in cellular components were nonmembrane-bound organelles, organelle parts and cell peripheries. The top three highly enriched processes in molecular functions were transmembrane transporter activity, transferase activity and hydrolase activity. According to a Kyoto Encyclopedia of Genes and Genomes (KEGG) analysis, the plant hormone signal transduction pathways were the most significantly enriched (Figure 3). In addition, plant–pathogen interactions, phenylpropanoid biosynthesis and the mitogen-activated protein kinase (MAPK) signalling pathway were also significantly enriched. These results indicate that plant hormones might play important roles in the high tillering of the mtn1 mutant.

### 2.5. Regulatory Pathways of DEGs Related to Plant Hormones

To identify the contributions of hormone-mediated regulation to high tilling in the *mtn1* mutant, we compared the expression of genes in hormone-related pathways. There were 44 DEGs related to auxin (33 DEGs), GA (6 DEGs) and CTK (5 DEGs) (Figure 4A and Appendix A). A total of six auxin transport-related DEGs were found. Among them, *ABCB11* and *PIN9* were upregulated, and the *Auxin1/like aux 4* (*AUX1/LAX4*) and *ABCB19* genes were downregulated in the *mtn1* mutant (Appendix A). A total of 16 DEGs were annotated as *small auxin upregulated RNA* (*SAUR*) genes, 10 of which were upregulated in the *mtn1* mutant (Appendix A). A total of two *Gretchen hagen 3* genes (*GH3.8* and *GH3.12*) were upregulated in the *mtn1* mutant (Appendix A). These results suggest that auxin is one of the main hormones regulating high tillering in the *mtn1* mutant, and the polar transport of this hormone was highly active, while the signal transduction pathway was reactive.

In this study, one DEG related to *Gibberellin 2-oxidase 1* (*GA2ox1*) was upregulated in the *mtn1* mutant (Figure 4B, Appendix A). A total of five GA signal transduction pathway-related genes were differentially expressed (Figure 4B and Appendix A). The *chitin-inducible gibberellin-responsive protein* (*CIGR*) gene was upregulated in the *mtn1* mutant, and *Gibberellic acid-stimulated transcripts in Arabidopsis 4* (*GASA4*), *GA-stimulated transcript 1* (*GAST1*), *Basic helix-loop-helix 106* (*bHLH106*) and *GA-insensitive 1* (*GAI1*) were downregulated in the *mtn1* mutant.

In this study, a total of three DEGs related to CTK metabolism were significantly expressed (Figure 4C and Appendix A). The expression levels of *LOG6* and *CKX5* were upregulated, and the *ZOG3* gene was downregulated in the *mtn1* mutant. One DEG related to CK transport (*Purine nucleobase transmembrane transport 3*, *PUP3*) was downregulated, and one DEG related to CK signal transduction (*Ethylene-responsive transcription factor 4*, *CRF4*) was upregulated in the *mtn1* mutant compared with the WT (Appendix A).

### 2.6. Regulatory Pathways of DEGs Involved in Starch and Sucrose Metabolism

Sugars play essential roles in tillering, so we analysed DEGs involved in starch and sucrose metabolism. A total of 11 DEGs were annotated to starch and sucrose metabolism (Figure 5 and Appendix A). In this pathway, the expression levels of *fructokinase 1* (*FRK1*), *beta-glucosidase 5* (*βGLU5*), *βGLU14*, *trehalose-phosphate phosphatase 7* (*TPP7*) and *beta-amylase 3* (*βAMY3*) were upregulated in the mtn1 mutant. In contrast, the expression levels of *beta-1,3-glucosidase 1* (*βGLU1*), *glysosomal beta-glucosidase* (*GLUA*), *sucrose synthase 7* (*SUS7*), *alpha-trehalose-phosphate synthase 5* (*TPS5*) and *alpha-amylase isozyme 3* (*AMY3*) were downregulated in the *mtn1* mutant (Appendix A).

### 2.7. Confirmation of RNA-Seq Data

Eight of the DEGs (*PIN9*, *ABCB11*, *SAUR36*, *SAUR50*, *IAA14*, *GH3.12*, *GA2ox1*, *CKX5*) were randomly selected to confirm the reliability of the RNA-seq data (Appendix A). The results indicated that the expression levels of the genes evaluated by qRT-PCR were consistent with those evaluated by RNA-seq. Therefore, the results of the RNA-seq were reliable.

### 2.8. IAA, GA, CTK and Soluble Sugar Contents and Their Influence on the Tillering Number

The results showed that the IAA and GA contents in the *mtn1* mutant were obviously lower than those in the WT (Appendix A), and isopentenyl adenosine (IPA) and soluble sugar contents in the *mtn1* mutant were higher than those in the WT (Appendix A). The application of IAA and GA_3_ decreased the tillering number (Appendix A), while 6-BA and starch increased the tillering number of the *mtn1* mutant and WT (Appendix A).

### 2.9. Overexpression of EoABCB11 in Arabidopsis Affected Branch Number

To further explore gene function, the *EoABCB11* gene was transferred into wild-type Arabidopsis. A total of eight transgenic lines were acquired, and six lines showed similar phenotypes. Three of the six transgenic lines were selected to measure the branch number. Compared with the wild-type (WT) plants, the overexpressed transgenic lines showed higher branch numbers (Figure 6).

### 2.10. Identification of SNPs

A total of 80,384 and 8,2247 SNPs in *mtn1* and WT were identified, mainly including transitions and transversions (Table 2). Transitions accounted for 64.07% in *mtn1* and 64.24% in WT. Transversions accounted for 35.93% in *mtn1* and 35.76% in WT, among which, the most common base substitution was C/T, followed by A/G, while the rarest was A/T (Table 2).

## 3. Discussion

The molecular mechanism underlying tillering in centipedegrass is unknown. In this work, we studied a high-tillering mutant of centipedegrass that exhibited high tillering, fast bud outgrowth, short internodes and large seeds. The high-tillering phenotype of the mtn1 mutant was the result of fast bud outgrowth. We used RNA-seq to explore the internal reason for high tillering in the mtn1 mutant. The results showed that the DEGs involved in plant hormone signal transduction pathways and sugar metabolism were highly enriched, indicating that plant hormones and sugars likely play important roles in regulating tillering in centipedegrass.

### 3.1. Regulatory Role of Plant Hormones in Centipedegrass Tillering

Auxin plays an important role in plant growth and development [8]. The balance of auxin content controlled by auxin transporters is essential for tillering [8,29,30]. In this work, the expression levels of auxin efflux carriers related genes *ABCB11* and *PIN9* were upregulated, and the expression levels of the auxin influx carriers related genes *AUX1/LAX4* and *ABCB19* were downregulated in the *mtn1* mutant. Those results were similar to those of Hou et al. [8], who reported that the overexpression of *OsPIN9* increases tillering in rice. The same result was also found in Arabidopsis; the *atabcb19* mutant showed a low level of auxin and more branches [31]. In this study, we also found that the *GH3.8 gene*, which promotes the degradation of auxin, was highly expressed in the *mtn1* mutant. This result was in agreement with a previous study on rice, which indicated that the overexpression of the *OsGH3.8* gene reduces auxin content and increases tillering numbers [32]. IAA14 is a transcriptional repressor that can interact with ARF5 to negatively regulate auxin response gene expression [33]. In this study, the expression level of *IAA14* was upregulated, which was consistent with Liu et al. [34], who found that the overexpression of the *PtrIAA14* gene in Arabidopsis results in more branches. This result suggested that *IAA14* might have a positive effect on centipedegrass tillering. We speculated that these genes maintain the low level of auxin that promotes tillering in centipedegrass.

Gibberellin 2-oxidases (GA2oxs) are important GA metabolism enzymes that reduce the GA content [35,36,37]. In this work, the expression level of *GA2ox1* was upregulated in the *mtn1* mutant. This result was in agreement with two previous studies on bahiagrass and hybrid aspen, which indicated that the overexpression of *PnGA2ox1* and *AtGA2ox1* leads to more branches [38,39]. We inferred that the *mtn1* mutant may accumulate less GA relative to *GA2ox1*, which might underlie the differences in tillering ability between the *mtn1* mutant and wild-type. Therefore, the overexpression of *GA2oxs* is a clear way to reduce GA levels and improve tillering in centipedegrass.

CTKs exist widely in plant tissues, directly promoting tillering by activating axillary bud development [40,41,42]. In this study, we found that the *LOG6* gene, which promotes the accumulation of CTK, was highly expressed. This result was consistent with Kurakawa et al. [43], who reported that the *OsLOG1* gene is required to maintain meristem activity and that its loss of function causes low tillering in rice. Otherwise, we found that the expression level of the *ZOG1* gene, which regulates cytokinin glucosylation levels, was downregulated in the *mtn1* mutant. The result was consistent with Shang et al. [18], who found that the knockdown of *OsZOG1* expression improves the tillering of rice. Taken together, these data suggested that CTK may act as a positive regulator of high tillering in centipedegrass, and the *mtn1* mutant accumulated more CTK, mainly as a consequence of the high expression level of *LOG6* and low expression level of *ZOG1*.

### 3.2. Regulatory Role of Sugars in Centipedegrass Tillering

In addition to plant hormones, sugars are also necessary for tillering, as they can act as an early signal and provide energy during the tillering development process [44,45,46]. In this work, the soluble sugar content in the *mtn1* mutant was higher than that of the wild-type. This result was in agreement with the previous opinion that plants with more tillering showed more sugar accumulation [46]. During the starch and sucrose metabolism process, *TPPs* catalyse the transformation of trehalose-6P into trehalose [23]. In this work, the expression level of *TPP7* was upregulated in the *mtn1* mutant. This result was in agreement with previous studies on Arabidopsis [24], maize [25] and upland cotton [45], which indicated that a high expression level of *TPPs* could increase the tillering/branch number. The high expression level of the *TPP7* gene was consistent with the high soluble sugar concentration and suggested that sugars are an energy source or nutrient aiding in bud growth in high-tillering centipedegrass.

Based on expression analysis of hormone and sugar-related genes, hormone and sugar contents and the results of previous studies, we propose a regulatory network model to explain the role of plant hormones and sugars in the control of high tillering in centipedegrass (Figure 7). We propose that plant hormones and sugars play primary regulatory roles in centipedegrass tillering. IAA and GA are negative regulators, and CTK is the positive regulator. The biosynthesis, transport and signal transduction of IAA, GA and CTK are required for fast cell expansion, division and tissue differentiation in high tillering. The increase in soluble sugar content provides a greater energy source, which improves protein and cell wall synthesis in the tillering of the *mtn1* mutant. The interaction between the hormones and sugars results in the tillering of the *mtn1* mutant.

### 3.3. EoABCB11 Was Involved in the Tillering of Centipedegrass

In total, four pathways were identified and examined in detail, revealing that centipedegrass tillering could be regulated by complex mechanisms mediated by plant hormones and/or sugar metabolism. Among these factors, we chose the IAA transport protein ABCB11 for further study, as the tillering number can be increased by increases in IAA auxin efflux. In this study, the branch number was increased by the overexpression of the *EoABCB11* gene in Arabidopsis. Similar results were found in Arabidopsis, which indicated that *atabcb19* and *atabcb1abcb19* mutants showed high branches [28,31]. However, the role of the *ABCB11* gene in tillering is still unknown. The ABCB protein can interact with PIN proteins to affect the stability of PIN in the plasma membrane region, thus improving the substrate specificity of the transporter. Previous studies found that ABCB1 and ABCB19 can interact with PIN1 and PIN2, and plants expressing *PIN1-ABCB1* showed an increase in auxin output [27,47]. In this study, consistent with *ABCB11*, the expression level of *PIN9* was upregulated, indicating that *EoABCB11* and *PIN9* might have an interactive relationship to regulate the tillering of centipedegrass. The molecular mechanism of *EoABCB11* and its governing effect on centipedegrass tillering requires further study.

### 3.4. SNPs Identification from mtn1 Mutant and WT

SNPs developed from RNA-seq have been widely used as powerful molecular markers for breeding and genetic research on plants [48,49]. In this study, SNPs were in agreement with previous studies, which indicated that the high content of A/G and C/T transitions in Chinese cabbage and cucumber [48,49]. The results provide a useful resource for future studies on gene function and breeding in centipedegrass.

## 4. Materials and Methods

### 4.1. Plant Materials

The centipedegrass cultivar ‘Yuxi’ (wild-type, WT) was got from the Institute of Botany in Jiangsu Province and the Chinese Academy of Sciences, Nanjing, China. A *more tillering number* (*mtn1*) mutant was obtained from calli of the WT radiated by cold plasma, a new radiation method.

### 4.2. Experimental Design

Experiments were performed at the Institute of Botany in Jiangsu Province and the Chinese Academy of Sciences, Nanjing, China (118°46′ E, 32°03′ N). Uniform stolons of WT and *mtn1* with the first three nodes were planted in a polyethylene plastic pot (35 cm in height and 30 cm in diameter) to investigate phenotypic and yield characteristics. Each pot was filled with 5 kg of soil:sand mixture (*w*/*w*, 4:1). The physical and chemical properties of the soil were: pH, 6.20; total nitrogen, 0.44 g kg^−1^; available phosphorus, 4.2 mg kg^−1^; available potassium, 137.01 mg kg^−1^; and dissolved organic carbon, 610.1 mg kg^−1^. Each pot contained one stolon. The pots were placed in a greenhouse with a rain shelter, and the temperature and humidity varied from 25–35 °C and 65–80%, respectively. Each type of material was planted in 20 pots. Ten pots of plants were used for phenotype and RNA-seq analysis, and another ten pots were used for yield analysis. Tillering phenotype characteristics, including tiller bud number and tillering bud length, were measured after 20 d of planting, and other characteristics were measured after 40 d of planting. Tiller buds of the *mtn1* mutant and wild-type were collected after 40 d of planting, frozen in liquid nitrogen and stored at −80°C for transcriptome and hormone analyses.

### 4.3. Phenotypic Characteristics and Yields

Tillering phenotype: Tiller bud number (a tiller bud length greater than 1 cm was considered a tiller) and tiller bud length were measured after 20 d of planting [15]. The tiller number, length and diameter of the third internode and the dry weights of the shoots and roots were measured after 40 d of planting. Ten individuals from each sample were randomly selected. Additionally, tiller buds (including nodes) with a length of 1–2 cm from the *mtn1* mutant and WT were collected separately as different samples with three replicates. The samples were frozen with liquid nitrogen and then stored at −80 °C for transcriptome and hormone analysis.

Yield characteristics: the length and width of seeds and 1000-seed weight were measured at the mature stage. Ten individuals were measured for each material.

### 4.4. RNA-Seq and Data Analyses

The total RNA of the *mtn1* mutant and WT was extracted using an RNAprep Pure Plant Kit (Takara, Japan). Six sequencing libraries were created using the NEBNext^®^ Ultra™ RNA Library Prep Kit for Illumina^®^ (NEB, Ipswich, MA, USA) according to the manufacturer’s instructions. Transcriptome sequencing was conducted at the Novogene Bioinformatics Institute (Beijing, China) on the Illumina NovaSeq 6000 platform (Illumina, San Diego, CA, USA). After removing low-quality reads, clean reads were mapped onto the centipedegrass reference genome [50] with HISAT2 (v2.0.5, http://daehwankimlab.github.io/hisat2/hisat-3n/, accessed on 17 July 2022) and were annotated with BLASTx (National Library of Medicine, Bethesda, MD, USA) [51]. The mapped reads of each sample were then assembled, and the novel transcripts were predicted using StringTie (v1.3.3b) (https://ccb.jhu.edu/software/stringtie/index.shtml, accessed on 17 July 2022). The expression levels of genes were calculated using fragments per kilobase per million fragments (FPKM). The DESeq2 R package (1.16.1) (http://bioconductor.org/packages/devel/bioc/vignettes/DESeq2/inst/doc/DESeq2.html, accessed on 17 July 2022) was used to analyse the differential expression level of genes between the two groups, with an adjusted padj <0.05 and |log2 (fold change)| ≥1. GO and KEGG pathway enrichment of the DEGs was analysed by the cluster Profiler R package (3.4.4) (http://www.r-project.org/, accessed on 17 July 2022) [52]. The metabolic and regulatory pathways were visualised by MapMan software (http://mapman.gabipd.org/, accessed on 19 November 2010). The RNA-Seq data from the three biological replicates were combined for SNP identification. SNPs were detected using SOAPsnp (Short Oligonucleotide Analysis Package) software (http://soap.genomics.org.cn/soapsnp.html, accessed on 17 July 2022). Then, the SNPs were identified through the consensus sequence in comparison with the reference sequence.

### 4.5. Quantitative Real-Time PCR Analysis of Expression of Candidate Genes

Based on our experimental results, we randomly selected 8 DEGs—*PIN9*, *ABCB11*, *SAUR36*, *SAUR50*, *IAA14*, *GH3.12*, *GA2ox1*, and *CKX5*—to verify the reliability of the transcriptome data. The total RNA was extracted from tiller buds of the WT and the mtn1 mutant using an RNAprep Pure Plant Kit (Takara, Japan). Primers for the 8 DEGs were designed using Primer 5.0 software (Appendix A). Quantitative real-time PCR (qRT-PCR) was carried out on the Bio-Rad CFX Connect PCR Detection System (Bio-Rad, Hercules, CA, USA) [53]. Each sample contained three biological replicates.

### 4.6. Analysis of Plant Hormones and Total Soluble Sugar Contents

IAA, GA, CTK and soluble sugar contents were measured in tiller buds (including nodes) of the WT and the *mtn1* mutant. A total of 0.50 g of each sample was used to measure IAA, cytokinin (IPA) and gibberellin acid (GA_1_ + GA_3_) contents, as described previously [54,55]. IAA, IPA, GA_1_ and GA_3_ were measured with high-performance liquid chromatography–electrospray ionization tandem mass spectrometry (HPLC-ESI-MS/MS; Agilent 1290 and TripleQuad 6500 HPLC/MS system, Agilent Technologies Inc., Santa Clara, CA, USA). Hormones were extracted with acetonitrile solution and purified using the QUECH method. Then, the solvent was concentrated with a nitrogen purge. Samples were subjected to HPLC-ESI-MS/MS analysis. Soluble sugar contents were measured through anthrone colourimetry [56]. Each sample consisted of three biological replicates.

### 4.7. Influence of IAA, GA, CTK and Soluble Sugar on Tillering Number

Uniform stolons of the WT and the *mtn1* mutant with the first three nodes were selected and planted in a bucket (20 cm in height and 15 cm in diameter). Each bucket contained 2.5 L of 1/2 Hoagland nutrient solution. The bucket was covered by a polyethylene plate with 14 spaced holes, and one stolon was planted in each hole. Starting one week after planting, the plant leaves were sprayed with 10 µM IAA, 25 µM GA_3_, 10 µM 6-benzylaminopurine (6-BA) and 1 µM sucrose once a week. Control plants were treated with distilled water. Buckets were placed in a greenhouse with a rain shelter. The nutrient solution was changed once a week. After three weeks, the tillering number was measured. Experiments were conducted following a completely randomised design with three replicates.

### 4.8. EoABCB11 Genetic Transformation

Based on the sequencing of the centipedegrass transcriptome, the *EoABCB11*-F/R primer pair (Appendix A) was designed using the Primer 5.0 software, and then the coding sequence (CDS) was amplified. The CDS of *EoABCB11* was first cloned into a pBI121 vector, and the vector was transformed into the Agrobacterium tumefaciens strain EHA105. The transgenic plants were produced via the floral dip method [57]. The primers used to detect the overexpression of transgenic plants were Bar-F/R (Appendix A). Homozygous T_3_ progeny were used for branch analysis. The branch numbers of the transgenic lines and the wild-type (each line contained five plants) were analysed.

### 4.9. Data Analysis

The morphological and hormone data are presented as the means ± standard errors. SPSS statistical software (version 16.0, IBM, Armonk NY, USA) and Origin 8.2 software (OriginLab, Northampton, MA, USA) were used to perform the statistical analyses. The variance (*p* < 0.05) of the data was analysed by one-way ANOVA (Sage Publications Inc., Newbury Park, CA, USA) (Duncan’s test).

## 5. Conclusions

Overall, the *mtn1* mutant exhibited high tillering, short internodes, long seeds and a heavy 1000-seed weight. The transcriptome results revealed that plant hormone signal transduction and starch and sucrose metabolism have interactive effects that contribute to the difference in tillering between the *mtn1* mutant and the WT. ABCB transporters might be involved in auxin transport to regulate tillering development. The transgenic results showed that overexpressing *EoABCB11* in Arabidopsis increases the branch number. The results of this study provide essential information for future studies of specific genes regulating tillering in centipedegrass and other perennial plants.

## Figures and Tables

**Figure 1 ijms-23-11580-f001:**
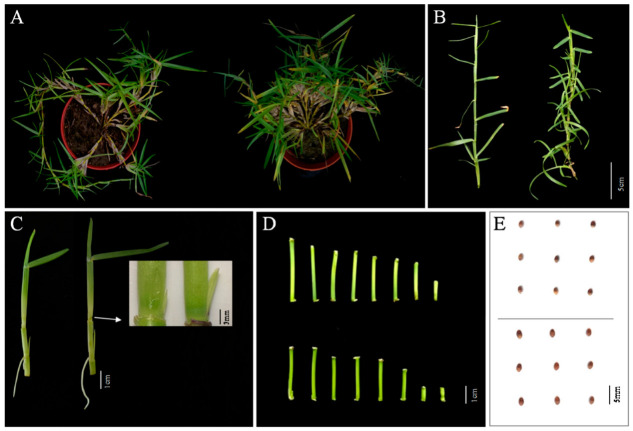
Comparison of the plant phenotype (**A**), stolon (**B**), tiller bud (**C**), internode (**D**) and seed (**E**) between the mtn1 mutant (**right**) and WT (**left**).

**Figure 2 ijms-23-11580-f002:**
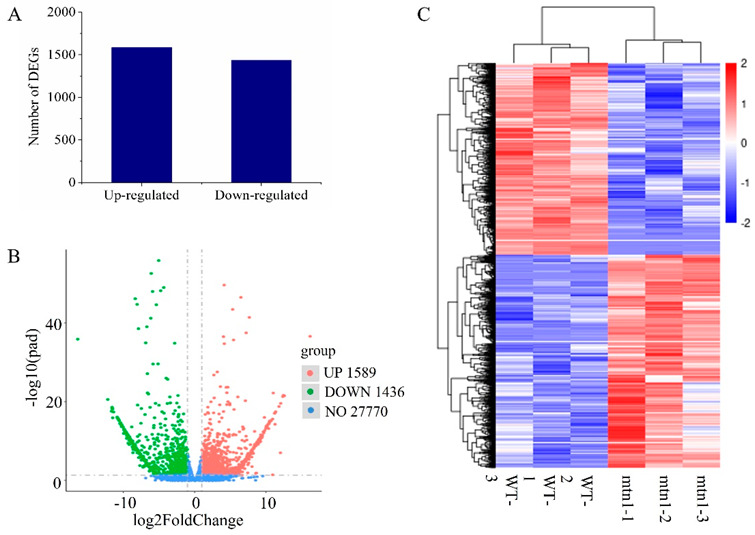
Expression levels of DEGs in the mtn1 mutant compared with those in the WT. (**A**) His togram chart of DEGs. (**B**) Volcano plots of DEGs. (**C**) Heatmap of DEGs. Colours from blue to red represent gene expression intensity, ranging from low to high.

**Figure 3 ijms-23-11580-f003:**
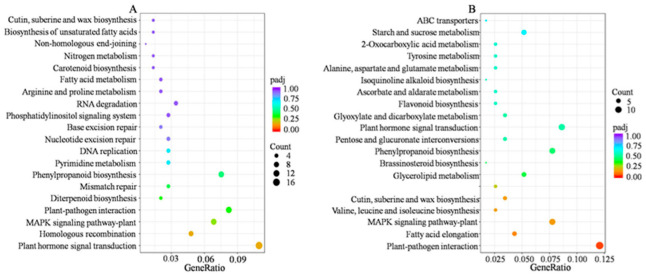
Top 20 KEGG classifications of upregulated DEGs (**A**) and downregulated DEGs between the mtn1 mutant and WT (**B**). High and low p values are represented by blue and red, respectively.

**Figure 4 ijms-23-11580-f004:**
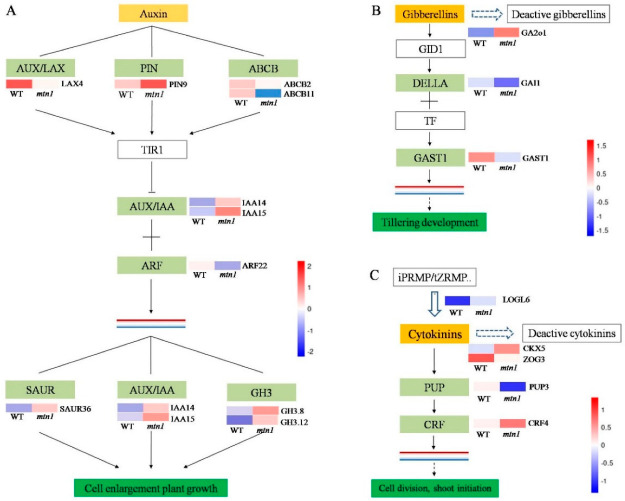
DEGs involved in plant hormone metabolism and signal transduction pathways. (**A**) Pathway in auxin polar transport and signal transduction, (**B**) Pathway in GA metabolism and signal transduction, (**C**) Pathway in CTK metabolism and signal transduction. The expression data are the TPM values of the samples; red indicates upregulated expression, and blue indicates downregulated expression.

**Figure 5 ijms-23-11580-f005:**
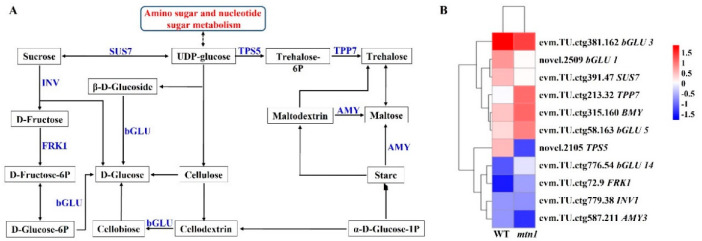
DEGs involved in starch and sucrose metabolism. (**A**) Pathway in starch and sucrose metabolism, (**B**) Heatmap of DEGs involved in starch and sucrose metabolism. The expression data are the TPM values of the samples; red indicates upregulated expression, and blue indicates downregulated expression.

**Figure 6 ijms-23-11580-f006:**
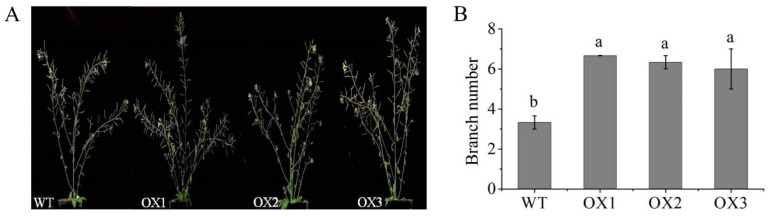
Overexpression of *EoABCB11* in Arabidopsis. (**A**) Branch phenotypes of WT and *EoABCB11*-overexpressing transgenic plants. (**B**) Total branch numbers of WT and *EoABCB11*-overexpressing transgenic plants. Different lowercase letters (a and b) above bars indicate significant differences between different plant types at the 0.05 level of significance (*p* < 0.05).

**Figure 7 ijms-23-11580-f007:**
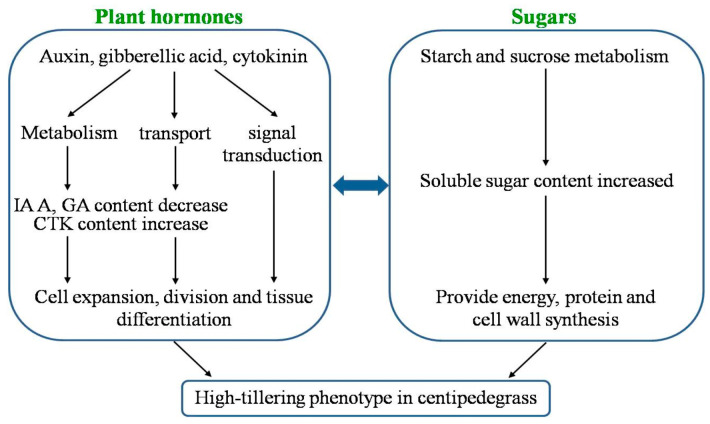
A possible model of the interaction between IAA, GA, CTK and sugars in the regulation of tillering.

**Table 1 ijms-23-11580-t001:** Comparison of phenotypic characteristics between the *mtn1* mutant and WT.

Traits	WT	*mtn1*
The primary tillering number	1.33 ± 0.33 b	2.50 ± 0.33 a
The second tillering number	9.00 ± 0.58 b	20.33 ± 1.20 a
Tiller bud number	1.10 ± 0.10 a	1.20 ± 0.13 a
First node of tiller bud length (mm)	1.02 ± 0.09 b	1.41 ± 0.13 a
Second node of tiller bud length (mm)	6.30 ± 0.64 b	11.12 ± 0.72 a
Third node of tiller bud length (mm)	12.94 ± 0.68 b	19.17 ± 0.85 a
Internode length (cm)	2.68 ± 0.085 a	1.82 ± 0.225 b
Internode diameter (mm)	2.17 ± 0.055 a	2.12 ± 0.027 a
Shoot dry weight (mg)	203.70 ± 10.72 b	362.38 ± 12.27 a
Root dry weight (mg)	77.00 ± 3.68 b	102.70 ± 6.74 a
Seed length (mm)	1.93 ± 0075 b	2.27 ± 0.023 a
Seed width (mm)	1.16 ± 0.014 a	1.19 ± 0.026 a
1000-seed weight (g)	0.99 ± 0.012 b	1.25 ± 0.015 a

Different lowercase letters in the same row indicate significant differences between different plant types at the 0.05 level of significance (*p* < 0.05).

**Table 2 ijms-23-11580-t002:** The SNPs types identified in *mtn1* mutant and wild type.

SNP Type	mtn1	WT
Transition		
C/T	26,436 (32.89%)	27,274 (33.16%)
A/G	25,066 (31.18%)	25,558 (31.07%)
Transversion		
C/G	8275 (10.30%)	8429 (10.25%)
G/T	7370 (9.17%)	7578 (9.21%)
A/C	7331 (9.12%)	7400 (9.00%)
A/T	5905 (7.35%)	6007 (7.30%)
Total	80,384	82,247

## Data Availability

The RNA-Seq datasets generated and analysed during this study are available in the NCBI Gene Expression Omnibus (GEO) repository, accession PRJNA863393. All other data generated during this study are included in this published article and its Appendix A.

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
