# Peer review of "Physiological and Comparative Transcriptome Analyses of the High-Tillering Mutant mtn1 Reveal Regulatory Mechanisms in the Tillering of Centipedegrass (Eremochloa ophiuroides (Munro) Hack.)"

_ijms, 2022, doi:10.3390/ijms231911580_

Round 1

Reviewer 1 Report

Tillering is a key factor determining centipedegrass's reproductive yields, an important perennial warm-season turfgrass. However, the regulatory mechanism of tillering in perennial plants is poorly understood, especially in perennial turfgrasses. We created and characterized a cold plasma-mutagenized centipedegrass mutant, mtn1 (more tillering number 1), in this study. Phenotypic analysis showed that the mtn1 mutant exhibited high tillering, short internodes, long seeds, and heavy 1000-seed weight. Then, comparative transcriptomic analysis of the mtn1 mutant 16 and wild type was used to explore the molecular mechanisms of tillering of centipedegrass. The results revealed that plant hormone signaling pathways and starch and sucrose metabolism might play important roles in centipedegrass tillering. Measurements of hormone and soluble sugar content and exogenous treatment results validated that plant hormones and sugars play important roles in centipedegrass tiller development. In particular, overexpression of the auxin transporter EoABCB11 in Arabidopsis resulted in more branches. According to the related genes' physiological characteristics and transcriptional expression levels, the regulatory mechanism of centipedegrass tillering was systematically revealed. This research provides a new breeding resource for studies into the molecular mechanism that regulates tillering in perennial plants and for breeding high-tillering centipedegrass varieties. 

This research is a very interesting topic and has a need to publish the results for breeding high tillering, eventually for yield. But it needs major reviews as below:

  1. Confused introduction with discussion. The discussion section is a combination of the introduction and results. The structure for each discussion point is: the first half is a repeated introduction, and the 2nd half is the repeated results. Discussions should compare your results with other reports, mainly discussing similarities and differences, and your opinions and hypotheses, if any. The discussion section needs to be completely rewritten. 
  2. The conclusion section is a mixture of conclusion with discussion. The proposed model is a hypothesis, NOT a conclusion. It needs to be placed in either result or discussion section. Consider shortening the conclusion to show the take-home messages. 
  3. There is no experiment design for this research, and it needs to be added for others to repeat the experiment, if any.
  4. For breeding purposes, the manuscript will be even better if single nucleotide polymorphism (SNP) can be developed from the main DEGs.
  5. There are several acronyms without the full words across the manuscript, and it needs to define when you use them the first time. 

Minor edits needed:

  1. In particular, overexpression of the auxin transporter 21 EoABCB11 in Arabidopsis resulted in more branches. This sentence in the abstract does not fit the overall story, and please consider revising it. 
  2. Several studies 44 have found that GA increases the branch number of sweet cherry [13] and Jatropha curcas 45 [14]. While, other studies found that GA inhibited the tillering/branching of tall fescue 46 [15], orchardgrass [16] and tomato [17]. The concentration of GA may be a factor, and please cite the GA concentration from the studies. 
  3. The ATP binding cassette B (ABCB) subfamily is the second largest ATP-binding cassette (ABC) protein subfamily is involved in the transmembrane transport of hormones, 59 metals, iron, etc. This sentence is not clear to the reviewer, and please rewrite it. 
  4. Finally, the ABCB 69 transporter EoABCB11 was selected from the DEGs. Whenever you use some acronyms first, define them. Here DEG needs to be defined. 

Author Response

Responses to Reviewer 1

Dear reviewer:

Thanks very much for your good comments on this manuscript. In the revision, necessary changes were made point by point in accordance with the reviewer’s comments and suggestions. First, we have rewritten the discussion and conclusion section. Second, we add the experiment design and SNPs analysis. In the end, we have made further modifications to the full manuscript, as suggested by reviewer.

Detailed remarks:

  1. Confused introduction with discussion. The discussion section is a combination of the introduction and results. The structure for each discussion point is: the first half is a repeated introduction, and the 2nd half is the repeated results. Discussions should compare your results with other reports, mainly discussing similarities and differences, and your opinions and hypotheses, if any. The discussion section needs to be completely rewritten.

Response: Thank you very much for this comment and good suggestion. I have rewritten this discussion section and highlighted changes in the revised version.

  1. The conclusion section is a mixture of conclusion with discussion. The proposed model is a hypothesis, NOT a conclusion. It needs to be placed in either result or discussion section. Consider shortening the conclusion to show the take-home messages.

Response: Thank you very much for this excellent suggestion. I have placed the model in discussion section, and shorten the conclusion section as you suggested. I have highlighted changes in the revised version.

  1. There is no experiment design for this research, and it needs to be added for others to repeat the experiment, if any.

Response: Thank you very much for this good comment. I have added the experiment design in the “4.2 Experiment design” section. Changes were highlighted in the revised version.

  1. For breeding purposes, the manuscript will be even better if single nucleotide polymorphism (SNP) can be developed from the main DEGs.

Response: Thanks for this good suggestion. I have added the single nucleotide polymorphism (SNP) data, this information was highlighted in the revised version.

  1. There are several acronyms without the full words across the manuscript, and it needs to define when you use them the first time.

Response: Thank you for this good suggestion. I have defined the acronyms with the full words and highlighted them in the revised version.

  1. In particular, overexpression of the auxin transporter EoABCB11 in Arabidopsis resulted in more branches. This sentence in the abstract does not fit the overall story, and please consider revising it. 

Response: Thank you for your good suggestion. I have revised this sentence into this: Among them, ATP binding cassette B (ABCB) subfamily participated in auxin hormone signal transduction pathways. Furthermore, an ABCB protein EoABCB11 overexpressed in Arabidopsis resulted in more branches.

  1. Several studies 44 have found that GA increases the branch number of sweet cherry [13] and Jatropha curcas 45 [14]. While, other studies found that GA inhibited the tillering/branching of tall fescue 46 [15], orchardgrass [16] and tomato [17]. The concentration of GA may be a factor, and please cite the GA concentration from the studies. 

Response: Thanks for your good comments. We found that high level of GA increased branch numbers in wood plants like sweet cherry (5000 mg/L) and Jatropha curcas (20 mM), while low level of GA decreased tillering number in tall fescue (25 μM), tomato (10-5 M) and rice (30 μM). So, the concentration of GA may not be a factor that influence branch/tillering. These findings demonstrated that GA might promote tillering/branch in perennial woody plants, but inhibit tillering/branch in annual herbaceous plants.

  1. The ATP binding cassette B (ABCB) subfamily is the second largest ATP-binding cassette (ABC) protein subfamily is involved in the transmembrane transport of hormones, metals, iron, etc. This sentence is not clear to the reviewer, and please rewrite it. 

Response: Thanks for your suggestion. I have rewritten it like this “ATP binding cassette B (ABCB) subfamily is the second largest ABC protein subfamily. They regulate the transport of plant hormones and heavy metals [27, 28]”.

  1. Finally, the ABCB transporter EoABCB11 was selected from the DEGs. Whenever you use some acronyms first, define them. Here DEG needs to be defined. 

Response: Thanks for your suggestion. I have defined the DEGs as differentially expressed genes (DEGs). I also defined other acronyms and highlighted them in the revised version.

Sincerely,

Ling Li

Jiangsu Province and Chinese Academy of Sciences

Zhongshanmenwai Qianhuahoucun, No. 1, Nanjing 210014, China

E-mail: liling7168@163.com

Reviewer 2 Report

Line40: Confirm the correct abbreviation for cytokinin (CTK).

L54: Correct "Trehelaose 6-phosphatase"

Fig 2A: Can you add std error or deviation? Are their any technical/biological replicates?

Figure 5: Higher resolution is needed.

L275: Delete "another"

L291: Correction " is involved" (not was involved)

L394: Delete sentence "Summary, we first compared the morphology between the mtn1 mutant and WT." and start with "Overall the..."

In general I recommend moderate language editing for conciseness and grammatical correctness.

Author Response

Responses to Reviewer 2

Dear reviewer:

Thanks very much for your good comments on this paper. In the revision, necessary changes were made point by point in accordance with the reviewer’s comments and suggestions.

Detailed remarks:

  1. Line40: Confirm the correct abbreviation for cytokinin (CTK).

Response: Thank you very much for your comment. I looked up a lot of books and articles and found that cytokinin could be abbreviated as either CTK or CK.

  1. L54: Correct "Trehelaose 6-phosphatase"

Response: Thank you very much for your suggestion. I have corrected Trehelaose 6-phosphatase in the paper and highlighted changes in the revised version.

  1. Fig 2A: Can you add std error or deviation? Are there any technical/biological replicates?

Response: Thanks for your suggestion. We did three biological replicates in the RNA-seq analysis. The number of differentially expressed genes (DEGs) was obtained by integrating three replicates. Usually, the plot representing the number of DEGs is shown without std error or deviation.

  1. Figure 5: Higher resolution is needed.

Response: Thanks for your suggestion. We have redrawn the Figure 5.

  1. L275: Delete "another"

Response: Thanks for your suggestion. We have deleted “another” and highlighted change in the revised version.

  1. L291: Correction " is involved" (not was involved)

Response: Thanks for your comment. We have corrected “is involved” and highlighted change in the revised version.

  1. L394: Delete sentence "Summary, we first compared the morphology between the mtn1 mutant and WT." and start with "Overall the..."

Response: Thanks for your comment. We have deleted sentence “"Summary, we first compared the morphology between the mtn1 mutant and WT” and start with “Overall

the mtn1 mutant exhibited high tillering, short internode length and long seed length, and a heavy 1000-seed weight”. We highlighted changes in the revised version.

  1. In general I recommend moderate language editing for conciseness and grammatical correctness.

Response: Thank you for your thoughtful comments. We have corrected the language of this manuscript, and the manuscript has been edited by a professional language editing company.

Sincerely,

Ling Li

Jiangsu Province and Chinese Academy of Sciences

Zhongshanmenwai Qianhuahoucun, No. 1, Nanjing 210014, China

E-mail: liling7168@163.com

Round 2

Reviewer 1 Report

The responses are goog enough for the acceptance.

Reviewer 2 Report

The authors have satisfactorily addressed most of the comments.